# Molecular Graph Representation Learning via Heterogeneous Motif Graph Construction

## Abstract

We consider feature representation learning problem of molecular graphs. Graph Neural Networks have been widely used in feature representation learning of molecular graphs. However, most existing methods deal with molecular graphs individually while neglecting their connections, such as motif-level relationships. We propose a novel molecular graph representation learning method by constructing a heterogeneous motif graph to address this issue. In particular, we build a heterogeneous motif graph that contains motif nodes and molecular nodes. Each motif node corresponds to a motif extracted from molecules. Then, we propose a Heterogeneous Motif Graph Neural Network (HM-GNN) to learn feature representations for each node in the heterogeneous motif graph. Our heterogeneous motif graph also enables effective multi-task learning, especially for small molecular datasets. To address the potential efficiency issue, we propose to use an edge sampler, which can significantly reduce computational resources usage. The experimental results show that our model consistently outperforms previous state-of-the-art models. Under multi-task settings, the promising performances of our methods on combined datasets shed light on a new learning paradigm for small molecular datasets. Finally, we show that our model achieves similar performances with significantly less computational resources by using our edge sampler.

## 1 Introduction

Graph neural networks (GNNs) have been proved to effectively solve various challenging tasks in graph embedding fields, such as node classification (Kipf & Welling, 2016), graph classification (Xu et al., 2018), and link prediction (Schlichtkrull et al., 2018), which have been extensively applied in social networks, molecular properties prediction, natural language processing, and other fields. Compared with hand-crafted features in the molecular properties prediction field, GNNs map a molecular graph into a dimensional Euclidean space using the topological information among the nodes in the graph (Scarselli et al., 2008). Most existing GNNs use the basic molecular graphs topology to obtain structural information through neighborhood feature aggregation and pooling methods (Kipf & Welling, 2016; Ying et al., 2018; Gao & Ji, 2019). However, these methods fail to consider connections among molecular graphs, specifically the sharing of motif patterns in the molecular graph.

One of the critical differences between molecular graphs and other graph structures such as social network graphs and citation graphs is that motifs, which can be seen as common sub-graphs in molecular graphs have special meanings. For example, an edge in a molecule represents a bond, and a cycle represents a ring. One ground truth that has been widely used in explanation of GNNs is that carbon rings and NO2 groups tend to be mutagenic (Debnath et al., 1991). Thus, motifs deserve more attention when designing GNNs for motif-level feature representation learning. To this end, we propose a novel method to learn motif-level feature embedding for molecular graphs. We first extract motifs from molecular graphs and build a motif vocabulary containing all these motifs. Then we construct a heterogeneous motif graph containing all motif nodes and molecular nodes. We can apply GNNs to learn motif-level representations for each molecular graph based on the heterogeneous motif graph. The message passing scheme in a heterogeneous motif graph enables interaction between motifs and molecules, which helps exchange information between molecular graphs. The experimental results show that the learned motif-level embedding can dramatically

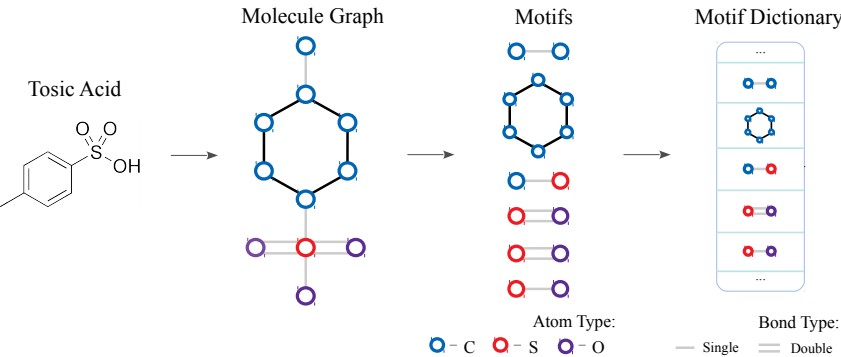

Figure 1: Example of building a motifs vocabulary. Given a molecule Tosic Acid, we first extract six bonds and rings from its atom graph. After removing duplicates, we add five unique motifs into the vocabulary. Blue, red, and purple nodes represent Carbon, Sulfur, and Oxygen atoms, respectively.

improve the representation of a molecule, and our model can significantly outperform other state-of-the-art GNN models on a variety of graph classification datasets.

## 2 HETEROGENEOUS MOTIF GRAPH NEURAL NETWORKS

In this section, we propose a novel method to construct a motif-based heterogeneous graph, which can advance motif-based feature representation learning on molecular graphs. Then, we use two separate graph neural networks to learn atom-level and motif-level graph feature representations, respectively.

### 2.1 MOTIF VOCABULARY OF MOLECULAR GRAPHS

In molecular graphs, motifs are sub-graphs that appear repeatedly and are statistically significant. Specific to biochemical molecule graphs, motifs can be bonds and rings. Analogously, an edge in the graph represents a bond, and a cycle represents a ring. Thus, we can construct a molecule from sub-graphs or motifs out of a motif vocabulary. To represent a molecule by motifs, we first build a motif vocabulary that contains valid sub-graphs from given molecular graphs.

To build the motif vocabulary, we search all molecular graphs and extract important sub-graphs. In this work, we only keep bonds and rings to ensure a manageable vocabulary size. However, the algorithm can be easily extended to include different motif patterns. We then remove all duplicate bonds and rings. Some motifs may appear in most of molecules, which carry little information for molecule representation. To reduce the impact of these common motifs, we employ the term frequency–inverse document frequency (TF-IDF) algorithm (Ramos et al., 2003). In particular, term frequency measures the frequency of a motif in a molecule, and inverse document frequency refers to the number of molecules containing a motif. We average the TF-IDFs of those molecules that contain a motif as the TF-IDF value of the motif. By sorting the vocabulary by TF-IDF, we keep the most essential motifs as our final vocabulary. Figure 1 illustrates the procedure of building motifs vocabulary.

### 2.2 HETEROGENEOUS MOTIF GRAPH CONSTRUCTION

Based on the motif vocabulary, we build a heterogeneous graph that contains motif nodes and molecular nodes. In this graph, each motif node represents a motif in the vocabulary, and each molecular node is a molecule. Then, we build two types of edges between these nodes; those are motif-molecule edges and motif-motif edges. We add motif-molecule edges between a molecule node and motif nodes that represent its motifs. We add a motif-motif edge between two motifs if they share at least one atom in any molecule. In this way, we can build a heterogeneous graph containing all motifs in the vocabulary and all molecules connected by two kinds of edges. Appendix A contains detailed pseudocode of constructing a Heterogeneous Motif Graph.

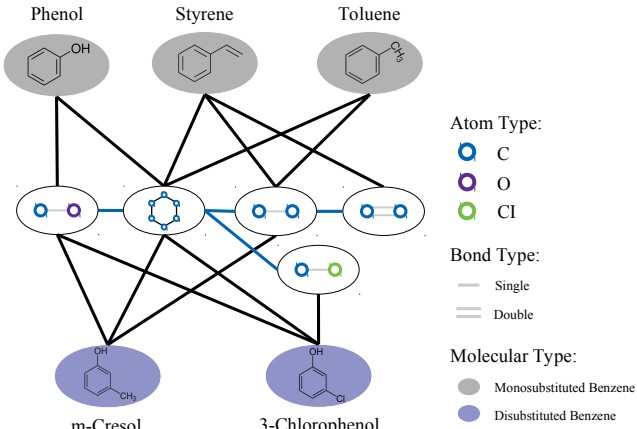

Figure 2: Example of a heterogeneous motif graph. In this graph, there are five molecular nodes: Phenol, Styrene, Toluene, m-Cresol, and 3-Chlorophenol. Here, we have five motifs in the vocabulary. We connect a molecular node with a motif node if the molecule contains this motif. For example, Phenol has Benzene and carbon–oxygen bond. Thus, we connect the Phenol node with the Benzene node and carbon-oxygen bond node. We connect two motif nodes if they share at least one atom in molecules. In this graph, we connect the Benzene node and the carbon-oxygen bond node since they share a carbon atom.

One thing to notice is that different motifs have different impacts. We assign different weights to edges based on their ending nodes. In particular, for edges between a motif node and a molecule node, we use the TF-IDF value of the motif as the weight. For the edges between two motif nodes, we use the co-occurrence information point-wise mutual information (PMI), which is a popular correlation measure in information theory and statistics (Yao et al., 2019). Formally, the edge weight $A_{ij}$ between node $i$ and node $j$ is computed as

$$A_{ij} = \begin{cases} \text{PMI}_{ij}, & \text{if } i, j \text{ are motifs} \\ \text{TF-IDF}_{ij}, & \text{if } i \text{ or } j \text{ is a motif} \\ 0, & \text{Otherwise} \end{cases} \tag{1}$$

The TF-IDF value of an edge between a motif node $i$ and a molecular node $j$ is computed as

$$\text{TF-IDF}_{ij} = C(i)_j \left( \log \frac{1+M}{1+N(i)} + 1 \right), \tag{2}$$

where $C(i)_j$ is the number of times that the motif $i$ appears in the molecule $j$, $M$ is the number of molecules, and $N(i)$ is the number of molecules containing motif $i$.

The PMI value of an edge between two motif nodes is computed as

$$\text{PMI}_{ij} = \log \frac{p(i,j)}{p(i)p(j)}, \tag{3}$$

where $p(i,j)$ is the probability that a molecule contains both motif $i$ and motif $j$, $p(i)$ is the probability that a molecule contains motif $i$, and $p(j)$ is the probability that a molecule contains motif $j$. We use following formulas to compute these probabilities.

$$p(i,j) = \frac{N(i,j)}{M}, \quad p(i) = \frac{N(i)}{M}, \quad p(j) = \frac{N(j)}{M}, \tag{4}$$

where $N(i,j)$ is the number of molecules that contain both motif $i$ and motif $j$. Figure 2 provides an example of heterogeneous motif graph construction. Note that we assign zero weight for motif node pairs with negative PMI value.

## 2.3 HETEROGENEOUS MOTIF GRAPH NEURAL NETWORKS

In this part, we build a HM-GNN to learn both atom-level and motif-level graph feature representations. In Section 2.2, we construct a heterogeneous motif graph that contains all motif nodes and

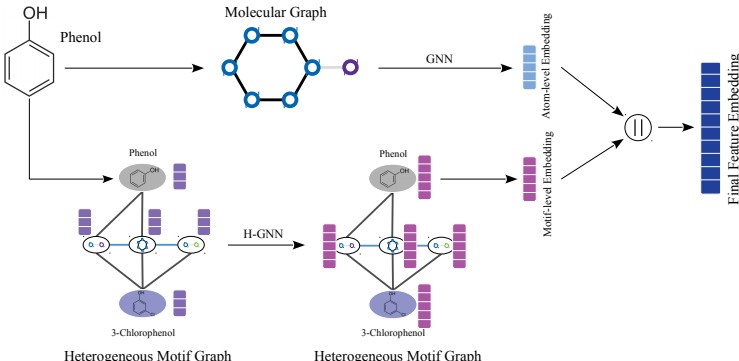

Figure 3: Example of our HM-GNN. Given an input Phenol, we first apply a GNN on its atom-level graph structure to learn its atom-level feature embedding. Meanwhile, we add it into a heterogeneous motif graph and use a heterogeneous GNN to learn its motif-level graph embedding. In the heterogeneous motif graph, Phenol is one of the molecular nodes. Finally, we concatenate graph embeddings from two GNNs and feed them into a MLP for prediction.

molecular nodes. Here, we first initiate features for each motif node and molecular node. We use the one-hot encoding to generate features for motif nodes. In particular, each motif node $i$ has a feature vector $X_i$ of length $|V|$, where $V$ represents the motif vocabulary we obtained in Section 2.1. Given the unique index $i$ of the motif in the vocabulary, we set $X_i[i] = 1$ and other positions to 0. For molecular nodes, we use a bag-of-words method to populate their feature vectors. We consider each motif as a word and each molecule as a document. By applying the bag-of-words model, we can obtain feature vectors for molecular nodes. Based on this heterogeneous graph, a heterogeneous graph neural network can be applied to learn the motif-level feature embedding for each molecule in the graph.

At the same time, each molecule can be easily converted into a graph by using atoms as nodes and bonds as edges. The original molecule graph topology and node features contain atom-level graph information, which can supplement motif-level information. Thus, we employ another graph neural network to learn the atom-level feature embedding. Finally, we concatenate feature embeddings from two graph neural networks and feed them into a multi-layer perceptron (MLP) for prediction. Figure 3 shows an example of our HM-GNN model.

## 2.4 MULTI-TASK LEARNING VIA HETEROGENEOUS MOTIF GRAPH

This part will show that our heterogeneous motif graph can help with graph deep learning models on small molecular datasets via multi-task learning. It is well known that deep learning methods require a significant amount of data for training. However, most molecular datasets are relatively small, and graph deep learning methods can easily over-fit on these datasets. Multi-task learning (Caruana, 1997) has been shown to effectively reduce the risk of over-fitting and help improve the generalization performances of all tasks (Zhang & Yang, 2017). It can effectively increase the size of train data and decrease the influence of data-dependent noise, which leads to a more robust model. However, it is hard to directly apply multi-task learning on several molecular datasets due to the lack of explicit connections among different datasets. Based on our heterogeneous motif graph, we can easily connect a set of molecular datasets and form a multi-task learning paradigm.

Given $N$ molecular datasets $D_1, \cdots, D_N$, each dataset $D_i$ contains $n_i$ molecules. We first construct a motif vocabulary $V$ that contains motifs from $N$ molecular datasets. Here, the motif only needs to be shared in some datasets but not all of them. Then, we build a heterogeneous motif graph that contains motifs in the vocabulary and molecules from all datasets. We employ our HM-GNN to learn both graph-level and motif-level feature representations for each molecule based on this graph. The resulting features of each dataset are fed into a separate MLP for prediction. In this process, the motif nodes can be considered as connectors connecting molecules from different datasets or tasks. Under the multi-task training paradigm, our motif-heterogeneous graph can improve the feature representation learning on all datasets.

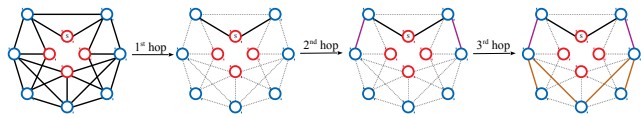

Figure 4: Example of generating a sub-graph for a 3-layer HM-GNN via an edge sampler. A sampling rule is we sample all edges, one edge, two edges for each layer, respectively (We select motif-motif edges at first). In this graph, we have four molecular nodes and seven motifs nodes. We use solid lines to represent selected edges and dashed lines to indicate unselected edges. We randomly choose node S as the "starting" node. In the first hop, we keep all edges connecting the node S and motif nodes. We sample one motif-motif edge for each motif node connecting to node S in the second hop. In the third hop, two motif-motif edges are selected for each newly added motif node in the last hop. Finally, the resulting sub-graph contains all nodes and edges we sampled.

### 2.5 EFFICIENT TRAINING VIA EDGE SAMPLING

In Section 2.2, we construct a heterogeneous motif graph that contains all motif nodes and molecular nodes. As the number of molecular nodes increases, there can be an issue with computational resources. To address this issue, we propose to use an edge sampler to reduce the size of the heterogeneous motif graph. Due to the special structure of our heterogeneous motif graph that it has two kinds of nodes and two kinds of edges, we can efficiently generate a computational subgraph by using the type of an edge.

We show how sampling edges can save computational resources. Most GNNs follow a neighborhood aggregation learning scheme. Formally, the $\ell$-th layer of a GNN can be represented by

$$x_i^{\ell+1} = f\left(x_i^\ell, \phi\left(\left\{e_{ji}, x_j^\ell \mid j \in \mathbb{N}(i)\right\}\right)\right), \tag{5}$$

where $x_i^{\ell+1}$ is the new feature vector of node $i$, $f$ is a function that combines the $\ell$-th layer's features and the aggregated features, $\phi$ is the function that aggregate all neighbors' feature vectors of node $i$, $e_{ji}$ is the weight of edge$_{ji}$, and $\mathbb{N}(i)$ is the set of node $i$'s neighbors. This equation shows that the time and space complexity are both $O(|E|)$, where $|E|$ is the number of edges in the graph. This means we can reduce the usage of computational resources by removing some edges from the graph. Thus, we employ an edge sampler that samples edges from the graph. And a sampling rule is that we prioritize motif-molecule edges.

To sample edges, we first randomly select some molecular nodes as "starting" nodes. We run a breadth-first algorithm to conduct a hop-by-hop exploration on the heterogeneous motif graph starting from these nodes. In each hop, we randomly sample a fixed size of edges based on edge type. Note that the first-hop neighbors of each molecular node are the motif nodes, which play essential roles in our heterogeneous motif graph. Thus, we retain all first-hop edges to ensure effective learning of feature representations for motif nodes. Starting from the second hop, we only sample motif-motif edges to retain as much motif information as possible. Figure 4 shows an example of sampling a sub-graph for a 3-layer HM-GNN. Appendix B contains complete sample rules and pseudocode.

## 3 EXPERIMENTAL STUDIES

In this section, we evaluate our proposed methods on graph classification tasks. We compare our methods with previous state-of-the-art models on various benchmark datasets. Datasets details and experiment settings are provided in the Appendix C and D.

### 3.1 PERFORMANCE STUDIES ON MOLECULAR GRAPH DATASETS

We first evaluate our model on five popular bioinformatics graph benchmark datasets from TU-Dataset (Morris et al., 2020), which includes four molecule datasets and one protein dataset. Here, PTC, MUTAG, NCI1, and Mutagenicity are molecule datasets, and PROTEINS is a protein dataset. Following GIN (Xu et al., 2018), we use node labels provided by TUDataset as the initial node features. To learn graph feature representations in our heterogeneous motif graphs, we use a 3-layer

Table 1: Graph classification accuracy (%) on various TUD graph classification tasks. Some results for GraphSAGE and GCN are reported from (Xu et al., 2018) and (Zhang et al., 2019). The best performer on each dataset are shown in **bold**. **-** means there is no reported accuracy on this dataset in original papers

.

| METHODS | PTC | MUTAG | NCI1 | PROTEINS | MUTAGENICITY |
|---|---|---|---|---|---|
| PatchySAN | $60.0 \pm 4.8$ | $92.6 \pm 4.2$ | $78.6 \pm 1.9$ | $75.9 \pm 2.8$ | - |
| GCN | $64.2 \pm 4.3$ | $85.6 \pm 5.8$ | $80.2 \pm 2.0$ | $76.0 \pm 3.2$ | $79.8 \pm 1.6$ |
| GraphSAGE | $63.9 \pm 7.7$ | $85.1 \pm 7.6$ | $77.7 \pm 1.5$ | $75.9 \pm 3.2$ | $78.8 \pm 1.2$ |
| DGCNN | $58.6 \pm 2.5$ | $85.8 \pm 1.7$ | $74.4 \pm 0.5$ | $75.5 \pm 0.9$ | - |
| GIN | $64.6 \pm 7.0$ | $89.4 \pm 5.6$ | $82.7 \pm 1.7$ | $76.2 \pm 2.8$ | - |
| PPGN | $66.2 \pm 6.5$ | $90.6 \pm 8.7$ | $83.2 \pm 1.1$ | $77.2 \pm 4.7$ | - |
| CapsGNN | - | $86.7 \pm 6.9$ | $78.4 \pm 1.6$ | $76.3 \pm 3.6$ | - |
| WEGL | $64.6 \pm 7.4$ | $88.3 \pm 5.1$ | $76.8 \pm 1.7$ | $76.1 \pm 3.3$ | - |
| GraphNorm | $64.9 \pm 7.5$ | $91.6 \pm 6.5$ | $81.4 \pm 2.4$ | $77.4 \pm 4.9$ | - |
| OURS | $\mathbf{78.8 \pm 6.5}$ | $\mathbf{96.3 \pm 2.6}$ | $\mathbf{83.6 \pm 1.5}$ | $\mathbf{79.9 \pm 3.1}$ | $\mathbf{83.0 \pm 1.1}$ |

GIN. To utilize atom-level information like node and edge features, we use another GIN on each atom-level graphs. Specifically, it has 5 GNN layers and 2-layer MLPs. Batch normalization (Ioffe & Szegedy, 2015) is applied to each layer, and dropout (Srivastava et al., 2014) is applied to all layers except the first layer. To evaluate the performance of our model, we strictly follow the settings in (Yanardag & Vishwanathan, 2015; Niepert et al., 2016; Xu et al., 2018; Gao & Ji, 2019). For each dataset, we perform 10-fold cross-validation with random splitting on the entire dataset. We report the mean and standard deviation of validation accuracy from ten folds.

We compare our model on five datasets with six state-of-the-art GNN models for graph classification tasks: PATCHY-SAN (Niepert et al., 2016), Graph Convolution Network (GCN) (Kipf & Welling, 2016), GraphSAGE (Hamilton et al., 2017), Deep Graph CNN (DGCNN) (Zhang et al., 2018), Graph Isomorphism Network (GIN) (Xu et al., 2018), Provably Powerful Graph Networks (PPGN) (Maron et al., 2019), Capsule Graph Neural Network (CapsGNN) (Xinyi & Chen, 2018), Wasserstein Embedding for Graph Learning (WEGL) (Kolouri et al., 2020), and GraphNorm (Cai et al., 2021). For baseline models, we report the accuracy from their original papers. The comparison results are summarized in Table 1. From Table 1, our model consistently outperforms baseline models on all five datasets. The superior performances on four molecular datasets demonstrate that the motif nodes constructed from the motif vocabulary can help GNN learn better motif-level feature representations of molecular graphs. On the protein dataset, our model also performs the best, which shows that the motifs in protein molecules also contain useful structural information.

## 3.2 Performance Studies on Large-Scale Datasets

To evaluate our methods on large-scale datasets, we use two bioinformatics datasets from the Open Graph Benchmark (OGB) (Hu et al., 2020): ogbg-molhiv and ogbg-molpcba. These two molecular property prediction datasets are adopted from the MOLECU-LENET (Wu et al., 2018). They are all pre-processed by RDKIT (Landrum et al., 2006). Each molecule has nine-dimensional node features and three-dimensional edge

Table 2: Graph Classification Results (%) on two open graph benchmark datasets. The results for GCN and GIN are reported from (Hu et al., 2020).

| METHODS | ogbg-molhiv | ogbg-pcba |
|---|---|---|
| GCN | $75.99 \pm 1.19$ | $24.24 \pm 0.34$ |
| GIN | $77.07 \pm 1.49$ | $27.03 \pm 0.23$ |
| PNA | $79.05 \pm 1.32$ | - |
| OURS | $\mathbf{79.03 \pm 0.92}$ | $\mathbf{28.70 \pm 0.26}$ |
| + PNA | $\mathbf{80.20 \pm 1.18}$ | - |

features. Ogbg-molhiv is a binary classification dataset, while Ogbg-molpcba is a multi-class classification dataset. We report the Receiver Operating Characteristic Area Under the Curve (ROC-AUC) for Ogbg-molhiv, and the Average Precision (AP) for Ogbg-molpcba, which are more popular for the situation of highly skewed class balance (only about 1.4% of data is positive). Following Wu et al. (2018); Hu et al. (2020), we adopt the scaffold splitting procedure to split the dataset. This evaluation scheme is more challenging, which requires the out-of-distribution generalization capability.

Table 4: Results on the PTC dataset with three different training settings. The first row report the performances of only using the PTC dataset. The second row and third row show the results of training on combined vocabularies and datasets with PTC_MM and PTC_FR, respectively. We report the motif vocabulary size (Vocab Size) of the dataset and the Overlap Ratio, which indicates the overlap ratio of motif vocabularies between two datasets. The last three columns represent the performances of using different sizes of training sets. For example, 90% means we use 90% of dataset as the training set and 10% of dataset as the testing set.

| Dataset | Vocab Size | Overlap Ratio | 90% | 50% | 10% |
|---|---|---|---|---|---|
| **PTC** | 97 | - | $71.8 \pm 4.1$ | $65.1 \pm 0.8$ | $59.9 \pm 1.9$ |
| **+ PTC_MM** | 111 | 83.5% | $76.5 \pm 3.3$ | $69.2 \pm 0.8$ | $66.7 \pm 1.9$ |
| **+ PTC_FR** | 110 | 94.8% | $84.3 \pm 3.8$ | $77.3 \pm 0.8$ | $74.0 \pm 1.7$ |

In this part, we compare our model with GIN (Xu et al., 2018), GCN (Kipf & Welling, 2016), and PNA (Corso et al., 2020). We report the mean and standard deviation of the results by using 10 different seeds (0-9). Table 2 shows the ROC-AUC results on Ogbg-molhiv and AP results on Ogbg-molpcba. It can be observed from the results that our approach outperforms GIN, GCN, and PNA by significant margins. The results demonstrate our model's superior generalization ability on large-scale datasets.

### 3.3 ABLATION STUDIES ON HETEROGENEOUS MOTIF GRAPH NEURAL NETWORKS

In Section 3.1 and Section 3.2, our HM-GNNs use GINs to learn atom-level information for molecular graphs, which can be a good complement to motif-level feature representations. To demonstrate the effectiveness of motif-level feature learning in our HM-GNNs, we remove the heterogeneous graphs and the corresponding GNNs from HM-GNNs, which reduces to GINs. We compare our HM-GNNs with GINs on three popular bioinformatics datasets: PTC, MUTAG, and PROTEINS. The comparison results are summarized in Table 3. From the results, our model significantly outperforms GIN by margins of 14.2%, 6.9%, and 3.7% on PTC, MUTAG, and PROTEINS datasets, respectively. This demonstrates that motif-level features are critical for molecular feature representation learning.

Table 3: Graph classification accuracy (%) of GIN and our model on three datasets: PTC, MUTAG, and PROTEINS.

| Models | PTC | MUTAG | PROTEINS |
|---|---|---|---|
| GIN | $64.6 \pm 7.0$ | $89.4 \pm 5.6$ | $76.2 \pm 2.8$ |
| OURS | $78.8 \pm 6.5$ | $96.3 \pm 2.6$ | $79.9 \pm 3.1$ |

### 3.4 RESULTS OF MULTI-TASK LEARNING ON SMALL MOLECULAR DATASETS

In the section 2.4, we introduce a new multi-task learning paradigm by constructing a mixed heterogeneous motif graph that contains molecular nodes from different datasets. Here, we conduct experiments to demonstrate the effectiveness of this new multi-task learning paradigm. To this end, we use another two PTC datasets: PTC_MM and PTC_FR. Here, both PTC and PTC_FR are rats datasets and PTC_MM is a mice dataset. By combining PTC with PTC_MM and PTC_FR separately, we can create another two datasets: PTC + PTC_MM and PTC + PTC_FR. The statistics of these new datasets are summarized in Table 4. In this table, we report the vocabulary sizes and the overlap ratios with the original PTC dataset. The overlap ratios show that these datasets share most of their motifs. In particular, the PTC dataset has 94.8% and 83.5% overlapped motifs with PTC_FR and PTC_MM, respectively. We construct separate heterogeneous motif graphs and evaluate our HM-GNNs on them. Here, we conduct the evaluations on three settings: 90%, 50%, and 10% for training and 10%, 50%, 90% for testing. The models trained on smaller training datasets have higher risk of overfitting, which can better reveal the impacts on performances.

The comparison results are summarized in the last three columns in Table 4. We can observe from the results that combing PTC with PTC_MM and PTC_FR can consistently bring performance improvements in three settings. Notably, combing PTC_FR brings even larger performance boost than using PTC_MM. This is because PTC has larger motif vocabulary overlap with PTC_FR. Thus, Combining datasets with similar motif vocabularies will benefit multi-task learning on small molecular datasets.

## 3.5 COMPUTATIONAL EFFICIENCY STUDY

In Section 2.5, we propose to use an edge sampler to improve the training efficiency of our HM-GNNs by reducing the size of edges in the heterogeneous graph. In our algorithm, if we fixed the sampling rule, the number of starting nodes is a hyper-parameter to control the size of the sampled heterogeneous motif graph. As an essential hyper-parameter, the number of starting nodes can influence both training efficiency and model performance. In this part, we conduct experiments to investigate its impact on the Ogbg-molhiv dataset. In particular, we change the number of starting nodes and report the corresponding model performances in terms of the ROC-AUC.

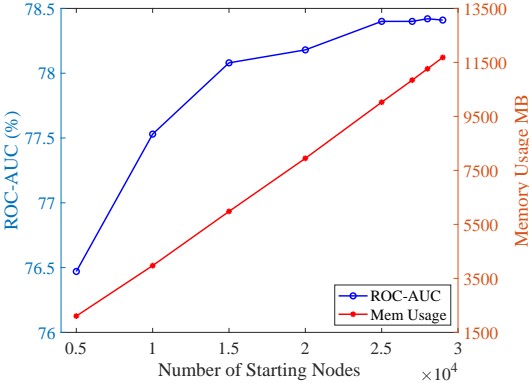

Figure 5: Results of ROC-AUC and memory usage using different number of starting nodes. We vary the number of starting nodes from 5,000 to 29,000. The ROC-AUC performances and memory usages of different batch sizes are illustrated on blue and red lines, respectively.

Figure 5 shows the performance of our model as the number of starting nodes changes. The blue line represents the ROC-AUC value, and the red line shows the memory usage. The blue line shows that the model performance boosts significantly when the number of starting nodes changes from 5,000 to 15,000. Starting from 15,000, the improvement of ROC-AUC gradually slows down until it converges. The red line shows that the memory usage increases almost linearly as the number of starting nodes increases. Thus, the model can achieve the best utility efficiency when choosing 25,000 as the number of starting nodes. At this point, our model can achieve high performance with relatively fewer computational resources.

## 3.6 MOTIF VOCABULARY SIZE STUDY

In Section 2.1, we propose filtering noisy motifs from the motif vocabulary by their TF-IDF values, which can improve our model's generalization ability and robustness. By using different keeping ratios, we can have different vocabulary sizes. In this part, we conduct experiments to investigate the impact of varying keeping ratios on the model performance. Using different keeping ratios, we can obtain different motif vocabularies on the Ogbg-molhiv dataset, leading to different heterogeneous motif graph construction outputs. Here, we apply our HM-GNNs on the resulting heterogeneous motif graphs and report the performances in terms of the ROC-AUC. Here, we vary the keeping ratio from 50% to 100%, which indicates the portion of top motifs using as final motif vocabulary.

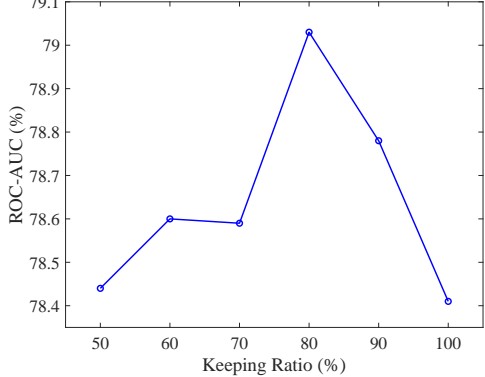

Figure 6: The impact of the motif keeping ratio. We evaluate our method using different keeping ratio of the original motif vocabulary.

We summarize the result in Figure 6. From the figure, we can observe that the model performance improves as the increase of keep ratios. A higher keeping ratio increases the motifs in the vocabulary, which leads to better motif-level feature propagation. And molecules in the graph have more connectors (motifs) to communicate with other molecules. The model performance starts to decrease when the keeping ratio is larger than 80%, which indicates the last 20% motifs are noisy and can hurt the model generalization and robustness. Notably, even with 50% of the most important motifs in the vocabulary, our model can still outperform GIN by a margin of 1.37%, which demonstrates the significant contribution of motif-level feature representations.

## 4 RELATED WORK

In this section, we provide a briefly introduction to some related works on graph representation, motif learning, and deep molecular graph learning.

GNNs (Micheli, 2009; Scarselli et al., 2008) have became the most major tool in machine learning on graph-related tasks. Many GNN variants have been proposed (Battaglia et al., 2016; Defferrard et al., 2016; Duvenaud et al., 2015; Hamilton et al., 2017; Kipf & Welling, 2016; Li et al., 2015; Veličković et al., 2017; Santoro et al., 2017; Xu et al., 2018). They have different neighborhood aggregation and graph pooling method. In factual, these models have achieved state-of-the-art baselines in many graph-related tasks like node classification, graph classification, and link prediction.

A motif can be a simple block of a complex graph and is highly related to the function of the graph (Alon, 2007). (Prill et al., 2005) demonstrates that dynamic properties are highly correlated with the relative abundance of network motifs in biological networks. MCN (Lee et al., 2019) proposes a weighted multi-hop motif adjacency matrix to capture higher-order neighborhoods and uses an attention mechanism to select neighbors. HierG2G (Jin et al., 2020) employs larger and more flexible motifs as basic building blocks to generate a hierarchical graph encoder-decoder. MICRO-Graph (Zhang et al., 2020) applies motif learning to help contrastive learning of GNN. Some GNN explainers also use motif knowledge to generate subgraphs to explain GNNs (Ying et al., 2019; Yuan et al., 2021).

Specific to the bioinformatics field, GNNs have been widely used in molecular graphs related tasks. Duvenaud et al. (2015) introduces a molecular feature extraction method based on the idea of circular fingerprints. MPNNs (Gilmer et al., 2017) reformulate existing models and explores additional novel variations. (Chen et al., 2017) introduces a non-backtracking operator defined on the line graph of edge adjacencies to enhance GNNs. DimeNet (Klicpera et al., 2020) proposes directional message passing to embed the message passing between atoms instead of the atoms themselves. HIMP (Fey et al., 2020) develops a method to learn associated junction trees of molecular graphs and exchange messages between junction tree representation and the original representation to detect cycles. Some other works introduce permutations of nodes into models (Murphy et al., 2019; Albooyeh et al., 2019). Pre-training and self-supervised learning schemes have been proved that they can effectively increase performance for downstream tasks (Hu et al., 2019; Sun et al., 2019; Rong et al., 2020; Hassani & Khasahmadi, 2020; Zhang et al., 2020; Sun et al., 2021). DGN (Beani et al., 2021) introduces a globally consistent anisotropic kernels for GNNs to overcome over-smoothing issue. Another line of work focuses on graph pooling. DiffPool (Ying et al., 2018) addresses a differentiable graph pooling module which can build hierarchical representations of graphs. Graph U-Nets (Gao & Ji, 2019) introduces pooling and up-sampling operations into graph data.

However, most existing works focus on the molecular graph itself, and they have not considered connections among molecular graphs, such as motif-level relationships. Because different molecules may share the same motif in their structures, we use a heterogeneous motif graph neural network model to extract and learn motif representation of molecular graphs. By learning motif-level feature representations, our proposed methods can increase the expressiveness of deep molecular graph representation learning.

## 5 CONCLUSION

In this work, we propose a novel heterogeneous motif graph and HM-GNNs for molecular graph representation learning. We construct a motif vocabulary that contains all motifs in molecular graphs. To remove the impact of noise motifs, we select the essential motifs with high TF-IDF values, which leads to more robust graph feature representations. Then, we build a heterogeneous motif graph that contains motif nodes and molecular nodes. We connect two molecules by jointly owned motifs in the heterogeneous motif graph, enabling message passing between molecular graphs. We use one HM-GNN to learn the heterogeneous motif graph and get the motif-level graph embedding. And we use another GNN to learn the original graph's atom-level graph embedding. This two-GNNs model can learn graph feature representation from both atom-level and motif-level. Experimental results demonstrate that our HM-GNN can significantly improve performance compared to previous state-of-the-art GNNs on graph classification tasks.

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

# A    PSEUDOCODE OF GENERATING A HETEROGENEOUS MOTIF GRAPH

Algorithm 1 is a pseudocode to show how to construct a Heterogeneous Motif Graph.
Lines 1-19 of Algorithm 1 generate all motif-molecule edges. Lines 20-24 build motif-motif edges. Line 25 first calculates TF-IDF for all molecules containing a motif, then averages these TF-IDFs as the TF-IDF value of the motif. Line 26 calculates the TF-IDF of a molecule as a molecule-motif weight, then calculates PMI value as a motif-motif weight. Note that we assign zero weight to motif node pairs with negative PMI. Lines 27-28 generate feature vectors for motif nodes and molecular nodes by one-hot encoding and bag-of-words method, respectively.

---

**Algorithm 1:** An algorithm of building a Heterogeneous Motif Graph

| | |
|---|---|
| **Input** | : A set of graph data G = $\{\mathcal{G}_1, \mathcal{G}_2, ..., \mathcal{G}_k, ..., \mathcal{G}_N\}, \forall k \in \{1, ..., N\}, n_k$ and $e_k$ denote the numbers of nodes and edges in graph $\mathcal{G}_k$, respectively. |
| **Output** | : A large Heterogeneous Motif graph $\mathcal{G}_M(\mathcal{V}, \mathcal{E}), \mathcal{V} = \mathcal{V}_{molecule} \cup \mathcal{V}_{motif},$ $\mathcal{E} = \mathcal{E}_{(molecule,motif)} \cup \mathcal{E}_{(motif,motif)}.$ |
| **Initialization:** | Initialize an empty set of motif vocabulary $M = \emptyset$ |
| | An empty Heterogeneous Motif graph $\mathcal{G}_M(\mathcal{V}, \mathcal{E}), \mathcal{V} = \emptyset, \mathcal{E} = \emptyset$ |
| | An empty dictionary $B$ stores how many times a motif appears in a molecule |
| | An empty dictionary $C$ stores how many molecules contain it for each motif. |

1  **for** $k = 1, ..., N$ **do**
2      Search Graph $\mathcal{G}_k(\mathcal{V}_k, \mathcal{E}_k)$
3      Generate a set of edges $M_E = \{e_1, e_2, ..., e_i, ..., e_I\}, \forall i \in \{1, ..., I\}$
4      and a set of basic cycles $M_C = \{c_1, c_2, ..., c_j, ..., c_J\}, \forall j \in \{1, ..., J\}$
5      **for** $e_i \in M_E$ **do**
6          **if** *$e_i$ is part of $c_j$ where $c_j \in M_C$* **then**
7              Delete $e_i$ from $M_E$
8          **end**
9      **end**
10     Unify motifs $M_k = M_E \cup M_C$
11     Add generated motifs into motif vocabulary $M = M \cup M_k$
12     Add a molecular node $v_k$ into $\mathcal{G}_M$
13     **for** *motif $s$ in $M_k$* **do**
14         Add a motif node $v_s$ into set $\mathcal{G}_M$
15         Add an edge between the motif node $v_s$ and the molecular node $v_k$ into set $\mathcal{E}$
16         $T$ donates how many $s$ does Graph $\mathcal{G}_k$ have and add $B_{ks} : T$ to $B$
17         Update $C_s$
18     **end**
19  **end**
20  **for** *motif $s$ and motif $r$ in $M$* **do**
21      **if** *$s$ and $r$ are sharing at least one atom* **then**
22          Add an edge between the motif node $s$ and the motif node $r$
23      **end**
24  **end**
25  Calculate TF-IDF for each motif, then select essential motifs based on some threshold.
26  Calculate molecule-motif edge weights based on dictionary $B$ and motif-motif edge weights based on $C$ and then add them to Graph $\mathcal{G}_M$
27  Generate one-hot feature vectors for motif nodes. $\{x \in \{0,1\}^p : \sum_{i=1}^p x_i = 1\}$, $p$ is the length of feature vector.
28  Generate feature vectors for molecular nodes using bag-of-word method.

---

## B  PSEUDOCODE OF OUR MINI-BATCH HETEROGENEOUS MOTIF GRAPH NEURAL NETWORK

Algorithm 2 is a pseudocode of minibatch HM-GNN. Lines 3-6 of Algorithm 2 correspond to the sampling stage. $\mathcal{R}^{(k)}$ is the sampling rule on each layer, and it fixes the sample size in each layer. To retain as much important motif information as possible, we sample all edges in the first hop, and starting from the second hop, we randomly sample edges from motif-motif edges. In this way, the computational graph contains essential motif nodes in sampling. Lines 7-12 correspond to a motif-level embedding learning stage. We can choose any kind of AGGREGATE$^{(k)}$· and COMBINE$^{(k)}(\cdot)$ here. Lines 14-19 correspond to an atom-level embedding learning stage. Line 21 concatenates the atom-level embedding and the motif-level embedding as a final graph embedding.

---

**Algorithm 2:** Mini-batch HM-GNN algorithm

| | |
|---|---|
| **Input** | : A set of graph data G = $\{\mathcal{G}_1, \mathcal{G}_2, ..., \mathcal{G}_k, ..., \mathcal{G}_N\}$, $\forall k \in \{1, ..., N\}$, $n_k$ and $e_k$ denote the number of nodes and edges |
| | Input features $\{x_g, \forall g \in G\}$ |
| | Heterogeneous Motif Graph $\mathcal{G}_M(\mathcal{V}, \mathcal{E})$, $\mathcal{V} = \mathcal{V}_{molecule} \cup \mathcal{V}_{motif}$ |
| | Input feature of Heterogeneous Graph $\{h_v^{(0)}, \forall v \in \mathcal{V}\}$ |
| | Depth K |
| | AGGREGATE$^{(k)}(\cdot)$ |
| | COMBINE$^{(k)}(\cdot)$ |
| | Edge sampling rule $\mathcal{R}^{(k)}$ |
| | Batch size $S$ |
| | A subset of G, we use this G′ to generate starting molecular nodes in HM graph sampling |
| | GNN model $p(\cdot)$ for atom-level learning |
| **Output** | : Vector representations $h$ for all molecular nodes in computational sampled graph |
| **Initialization:** | An empty computational graph for motif learning $\mathcal{G}_C(\mathcal{V}_C, \mathcal{E}_C)$, $\mathcal{V}_C = \emptyset$, $\mathcal{E}_C = \emptyset$ |

1 Generate starting molecular nodes $\mathcal{B}^{(K)} \leftarrow$ G′
2 Add all nodes in $\mathcal{B}^{(K)}$ into $\mathcal{V}_C$
3 **for** $k = K...1$ **do**
4      Randomly sample edges based on rule $\mathcal{R}^{(k)}$ and starting nodes $\mathcal{B}^{(k)}$ from $\mathcal{G}_M$, and then add these edges into $\mathcal{E}_C$, add all end nodes of edges into $\mathcal{V}_C$
5      All end nodes of these edges become a new set of starting nodes $\mathcal{B}^{(k-1)}$ for next iteration
6 **end**
    `// Learning computational graph` $\mathcal{G}_C$ `for motif embedding`
7 **for** $k = 1...K$ **do**
8      **for** $v \in \mathcal{B}^{(k)}$ **do**
9          $a_v^{(k)} = \text{AGGREGATE}^{(k)}\left(\{h_u^{(k-1)} : u \in \mathcal{N}(v)\}\right)$
10          $h_v^{(k)} = \text{COMBINE}^{(k)}\left(h_v^{(k-1)}, a_v^{(k)}\right)$
11      **end**
12 **end**
13 Motif embedding $e_m \leftarrow h_v, \forall v \in \mathcal{B}^{(K)}$
    `// Learning atom-level embedding`
14 **for** $v \in \mathcal{B}^{(K)}$ **do**
15      Find the corresponding atom-level graph $g$
16      $h_a = p(g, x_g)$
17 **end**
18 Atom-level embedding $e_a \leftarrow h_a, \forall a \in \mathcal{B}^{(K)}$
19 Final graph embedding $\mathcal{E} = e_a \| e_m$

---

## C  DETAILS OF DATASETS

We give detailed descriptions of datasets used in our experiments. Further details can be found in Yanardag & Vishwanathan (2015), Zhang et al. (2019), and Wu et al. (2018).

MUTAG (Debnath et al., 1991) is a dataset that contains 188 mutagenic aromatic and heteroaromatic nitro compounds. The task is to predict their mutagenicity on Salmonella typhimurium. It has 7 discrete labels. PTC (Toivonen et al., 2003) is a dataset that contains 344 chemical compounds that reports the carcinogenicity for male and female rats with 19 discrete labels. NCI1 (Wale et al., 2008) is made publicly available by the National Cancer Institute (NCI). It is a subset of a screened compound balance dataset designed to inhibit or inhibit the growth of a group of human tumor cell lines with 37 discrete labels. PROTEINS (Borgwardt et al., 2005) is a dataset in which nodes are secondary structure elements (SSE). If two nodes are adjacent nodes in an amino acid sequence or 3D space, there are edges between them. It has 3 discrete labels, which represent spirals, slices, or turns. Mutagenicity (Kazius et al., 2005) is a dataset of compounds for drugs, which can be divided into two categories: mutagens and non-mutagenic agents.

The Ogbg-molhiv dataset was introduced by the Drug Therapy Program (DTP) AIDS Antiviral Screen, which tested the ability of more than 40,000 compounds to inhibit HIV replication. The screening results were evaluated and divided into three categories: Confirmed Inactivity (CI), Confirmed Activity (CA), and Confirmed Moderate Activity (CM). PubChem BioAssay (PCBA) is a database consisting of the biological activities of small molecules produced by high-throughput screening. The Ogbg-pcba dataset is a subset of PCBA. It contains 128 bioassays and measures more than 400,000 compounds. Previous work was used to benchmark machine learning methods.

# D    DETAILS OF EXPERIMENT SETTINGS

We give detailed descriptions of experiment settings used in our experiments.

**TUDatasets.** For all configurations, 3 GNN layers are applied, and all MLPs have 2 layers. Batch normalization is applied on every hidden layer. Dropout is applied to all layers except the first layer. The batch size is set to 2000. We use Adam optimizer with initial weight decay 0.0005. The hyper-parameters we tune for each dataset are: (1) the learning rate $\in \{0.01, 0.05\}$; (2) the number of hidden units $\in \{16, 64, 1024\}$; (3) the dropout ratio $\in \{0.2, 0.5\}$.

**Open Graph Benchmark.** For all configurations, 3 GNN layers are applied, and all MLPs have 2 layers. Batch normalization is applied on every hidden layer. Dropout is applied to all layers except the first layer. We use Adam optimizer with initial weight decay 0.0005, and decay the learning rate by 0.5 every 20 epochs if validate performance not increase. The hyper-parameters we tune for each dataset are: (1) the learning rate $\in \{0.01, 0.001\}$; (2) the number of hidden units $\in \{10, 16\}$; (3) the dropout ratio $\in \{0.5, 0.7, 0.9\}$ (4) the batch size $\in \{128, 5000, 28000\}$.

