# OpenReview forum: "Molecular Graph Representation Learning via Heterogeneous Motif Graph Construction"
_ICLR.cc/2022/Conference — ICLR 2022 Submitted_

### Official Review · Reviewer_VxSx · 2021-10-30

**Correctness:** 3
**Technical Novelty And Significance:** 2
**Empirical Novelty And Significance:** 2
**Recommendation:** 5
**Confidence:** 4

**Main Review:**

Strengths:
1. The idea of constructing a motif-molecule heterogeneous graph is new;
2. The experimental results show that the proposed method can improve the performance significantly;
3. The paper is basically well-written and easy to follow.

Weaknesses:
1. Motif has been explored extensively in the literature, which makes the contribution of novelty for this paper is marginal;
2. Though the idea of constructing a motif-molecule heterogeneous graph is new, it seem unnecessary to introduce motif nodes in this graph;
3. The technical contribution of this paper includes three parts, but the three parts are not necessarily related to each other, which makes the paper very "loose" and more like a combination of small tricks;
4. The part of edge sampling seems trivial;
5. The authors do not test their method on some datasets that are commonly used for molecule property prediction, which makes it hard to comprehensively compare their method with the literature;
6. Some of the technical details are not clearly stated (see below for detail);
7. The quality of figures can be improved.


Details for weaknesses:
1. This paper studies the problem of using motifs (functional groups) in molecule representation learning. However, it is well known that molecule motif is an essential part of molecule structure, which have been extensively explored in the literature.
2. In the molecule-motif graph constructed by the authors, there are motif-motif links (co-occurrence relation) and motif-molecule links (part-whole relation), but no molecule-molecule links. However, motifs are more like a bridge connecting molecules in this big graph, and what the authors finally use is only molecule representation. If this is the case, it seems more direct and reasonable to build a molecule-molecule graph, whose edge weights are calculated based on the motifs they consists of, then the message can pass directly among molecules in GNN. I think the authors can at least take this as a baseline method? Moreover, it is also interesting to investigate if there is any theoretical relationship between the two graphs in terms of running GNN on them.
3. The technical contribution of this paper consists of three parts: constructing a molecule-motif graph, multi-task learning framework, and edge sampling. However, they are not necessarily related to each other. In other words, even if you do not use the molecule-motif graph, you can still build another molecule-molecule graph and use your multi-task and edge sampling strategy on top of this graph. The three parts of contributions are loosely coupled together, making this paper more like a combination of separate tricks.
4. The part of edge sampling seems trivial to me: of course you can randomly delete some edges from a graph to improve the time efficiency, but you do loss information from the original graph. Therefore, it is always a trade-off, which should be discussed in the paper. Moreover, it is also weird why the authors highlight a BFS method on sampling edges: It is easy to implement a method that directly removes edges from the graph while ensuring that all nodes have at least one edge left.
5. There are a lot of molecule property datasets proposed in the literature, for example, BBBP, BACE, HIV, Tox21, ToxCast, SIDER, QM9, ESOL, FreeSolv, etc, but the authors do not test their methods on these commonly used datasets.
6. Some of the technical details are not clearly stated: (1) In section 2.1 the authors say that "By sorting the vocabulary by TF-IDF, we keep the most essential motifs as our final vocabulary", but TF-IDF measures the importance of a motif to a specific molecule, and it is not an independent feature of motifs. How you get the TF-IDF value of motifs? Did you average the TF-IDF values of a motif to all molecules and treat this as the TF-IDF value of this motif? (2) In section 2.2, the authors construct a heterogeneous graph and define edge weights as in Eq. (1), however, PMI could be positive or negative, and TF-IDF can only be positive. How do you deal with negative PMI values in GNN? Should these two type of edge weights be normalized or calibrated to make sure that they are at the same scale in the unified graph?
7. Though the figures in this paper are clear and easy to read, I suggest that the authors reorganize (do not leave too blanks), resize (make sure that texts and shapes across these figures are basically of the same size), and recolor the graphs (do not use colors that are too bright, try to color your graphs using the same style) to make them more beautiful and professional.

**Summary Of The Paper:**

This paper proposes learning molecule representations by using motif-level information. The authors construct a heterogeneous graph which consists of molecules and motifs, then learn representations of them using graph neural networks. The learned features are concatenated with the molecule feature learned from a traditional atom-level graph neural networks and fed into an MLP for property prediction.

**Summary Of The Review:**

In summary, I think the idea of using motif in molecule representation learning is interesting, but the propose method has a lot of design issues that are not well-justified or well-explained. Moreover, the proposed method is not tested on commonly-used datasets. Overall, I think this paper is below the borderline and I tend to reject this paper.

---

> ### Author Response · Authors · 2021-11-16
> **Response to Reviewer VxSx (Part 1/2)**
>
> Thanks for your insightful review.
>
> **Concern 1:** This paper studies the problem of using motifs (functional groups) in molecule representation learning. However, it is well known that molecule motif is an essential part of molecule structure, which have been extensively explored in the literature.
>
> **Answer 1:** Most existing literature focuses on atom-level motif learning. For example, MICRO-Graph [1] applies motif learning to help contrastive learning. HierG2G [2] uses motifs to generate a graph encoder-decoder. In this work, we propose a method to learn motif information between graphs, which is motif-level embedding learning. We use motifs as connectors that connect molecular graphs, which is fundamentally different from previous motif-based learning methods. Also, our Heterogeneous Motif graph introduces a multi-task learning scheme to small dataset learning, which cannot be accomplished at atom-level.
>
> ***
>
> **Concern 2:** Though the idea of constructing a motif-molecule heterogeneous graph is new, it seem unnecessary to introduce motif nodes in this graph.
>
> **Answer 2:** Thank you for pointing this. We address your concern as follows:
>
> 2.1. In our paper, we claim that motif embedding is important for molecular representation. We expect motif embedding can be applied to other learning schemes like transfer learning and self-supervised learning. If we don’t have motif nodes in this graph, we cannot learn motif representation.
>
> 2.2. If we only have molecular nodes in a graph, the number of edges will greatly increase, and the graph will be dense. Take an extreme example, if n molecules are sharing one motif. Our graph will have n edges, while a graph only has molecular nodes will have $n^2-1$ edges. The graph will be dense, and It is difficult to learn useful information with a dense graph.
>
> We conducted an experiment on small datasets to compare the performances between our Heterogeneous Motif graph and Molecule-Molecule graph. Note that we do not have atom-level learning on this experiment because we want a model that mainly learns from the motif information. The result below shows that GNN can learn pretty good motif embedding from our Heterogeneous Motif graph but can not benefit from a Molecule-Molecule graph.
>
> | Method               |     PTC    |   MUTAG  |    NCI1    |  PROTEINS  | MUTAGENICITY |
> |----------------------|:----------:|:--------:|:----------:|:----------:|:------------:|
> | # Nodes              |     188    |    344   |    4110    |    1113    |     4337     |
> | # Edges in HM Graph  |     840    |   1365   |    23822   |    42811   |     24713    |
> | # Edges in M-M Graph |    17766   |   47895  |   8344704  |   565681   |    9342083   |
> | Accuracy (HM Graph)  | 73.8 ± 7.7 | 93.3±3.5 | 76.9 ± 1.8 | 78.5 ± 2.8 |  82.5 ± 1.3  |
> | Accuracy (M-M Graph) |  59.9±1.3  | 64.5±1.2 |  52.7±0.4  |  68.1±0.1  |   53.5±0.5   |
>
> ***
>
> **Concern 3:** The technical contribution of this paper includes three parts, but the three parts are not necessarily related to each other, which makes the paper very "loose" and more like a combination of small tricks.
>
> **Answer 3:** Previously, we have shown that a molecule-molecule graph is not a good choice since it leads to a much denser graph and increases the risk of overfitting.
>
> For the contributions of our work, they are closely related based on the construction and usage of a Heterogeneous Motif Graph. Multi-task learning is a promising application of our Heterogeneous Motif Graph, which can resolve the problem of small datasets in the community. The edge sampling strategy is to address the potential efficiency problem when using a Heterogeneous Motif Graph. Thus, we believe they are closely related. Multi-task learning and edge sampling are necessary to explore broader applications of our Heterogeneous Motif Graph.
>
> ***
>
> **Concern 4:** The part of edge sampling seems trivial.
>
> **Answer 4:** We have added some detailed explanations in section 2.5 and Appendix B. Specifically, we carefully designed our edge sampler to best fit our Heterogeneous Motif Graph. We sample all edges in the first hop and only sample motif-motif edges starting from the second hop. This sampling method can retain as much motif information as possible.

---

> > ### Comment · Reviewer_VxSx · 2021-12-01
> > **Thanks for the response**
> >
> > I have read the authors' rebuttal. Though the authors emphasized their contribution of this paper, I don't think the difference between their work and the literature using motifs is significant. Moreover, the authors did not clearly address my concern 3. For some of my concerns, the authors said that they will address them in the future, which makes the paper hard to meet the standard of ICLR based on its current form. Therefore, I would like to keep my score unchanged. Thanks!

---

> ### Author Response · Authors · 2021-11-16
> **Response to Reviewer VxSx (Part 2/2)**
>
> **Concern 5:** Moreover, it is also weird why the authors highlight a BFS method on sampling edges: It is easy to implement a method that directly removes edges from the graph while ensuring that all nodes have at least one edge left.
>
> **Answer 5:** Our heterogeneous motif graph has two kinds of nodes and edges, so we need to consider them differently. A molecular node needs motifs information to learn motif-level representation, and a motif node is a bridge between two molecular nodes. The strategy of random sampling without differentiation will lose too much motif information.
>
> ***
>
> **Concern 6:** There are a lot of molecule property datasets proposed in the literature, for example, BBBP, BACE, HIV, Tox21, ToxCast, SIDER, QM9, ESOL, FreeSolv, etc, but the authors do not test their methods on these commonly used datasets.
>
> **Answer 6:** Thank you for pointing out these molecule property datasets.
>
> We have tested our model on five popular small bioinformatic datasets and two large open graph benchmarks. Since conducting evaluations on these datasets is time-consuming, we will provide more results on them in the future.
>
> ***
>
> **Concern 7:** Some of the technical details are not clearly stated: (1) In section 2.1 the authors say that "By sorting the vocabulary by TF-IDF, we keep the most essential motifs as our final vocabulary", but TF-IDF measures the importance of a motif to a specific molecule, and it is not an independent feature of motifs. How you get the TF-IDF value of motifs? Did you average the TF-IDF values of a motif to all molecules and treat this as the TF-IDF value of this motif? (2) In section 2.2, the authors construct a heterogeneous graph and define edge weights as in Eq. (1), however, PMI could be positive or negative, and TF-IDF can only be positive. How do you deal with negative PMI values in GNN? Should these two type of edge weights be normalized or calibrated to make sure that they are at the same scale in the unified graph?
>
> **Answer 7:** Thank you for pointing out these details.
>
> For TF-IDF, we first calculate all molecules' TF-IDF. Then we average those TF-IDFs that are not equal to zero as the TF-IDF of motifs.
> For PMI, we set zero weight to edges have negative PMI. We add these details in the updated pdf file.
>
> ***
>
> **Concern 8:** Though the figures in this paper are clear and easy to read, I suggest that the authors reorganize (do not leave too blanks), resize (make sure that texts and shapes across these figures are basically of the same size), and recolor the graphs (do not use colors that are too bright, try to color your graphs using the same style) to make them more beautiful and professional.
>
> **Answer 8:** Thank you for giving these suggestions.
>
> We will improve our figures in the final revision.

---

### Official Review · Reviewer_6PpB · 2021-10-31

**Correctness:** 3
**Technical Novelty And Significance:** 3
**Empirical Novelty And Significance:** 3
**Recommendation:** 6
**Confidence:** 5

**Main Review:**

Pros
- The idea of adopting multi-task learning via heterogenous motif graph to address lack of data problem of deep learning for molecular data is interesting. Moreover, the empirical performance through multi-task learning is promising.
- The paper is well organized.

Cons
1. The paper lacks a clear justification on whether two molecules sharing motifs imply special meaning.
	- In other words, is sharing a motif a clear indication that two molecules share common properties?
	- In addition, a motif-motif edge is generated if at least one atom is shared. Does sharing an atom imply anything chemically?

2. The exact message passing process needs to be described. For example, the sizes of the feature vectors of a motif node and a molecular node should be different, and how they are put together for message passing is not clear.

3. Why are there blanks in Table 1 regarding "MUTAGENICITY."

4. Regarding the edge sampler,
	- Is BFS done until all the nodes are covered?
	- How efficient is HM-GNN compared with GIN in terms of memory consumption?
	- The edge sampler seems to be not much different from what is done in GraphSAGE.

5. The literature survey is not sufficient.
	- There are many recent work on GNNS for molecular graphs especially along the line of self-supervised learning, such as [1,2,3,4]. HM-GNN should be compared with a few methods among them, if not, they should be mentioned in the related work.
	- Moreover, since this is not the first paper to consider "motifs" for GNNs, related work, such as [5,6,7,8] should have been described in the paper.


[1] MoCL: Contrastive Learning on Molecular Graphs with Multi-level Domain Knowledge

[2] InfoGraph: Unsupervised and Semi-supervised Graph-Level Representation Learning via Mutual Information Maximization

[3] Contrastive Multi-View Representation Learning on Graphs

[4] Self-Supervised Graph Transformer on Large-Scale Molecular Data

[5] Graph Convolutional Networks with Motif-based Attention

[6] Hierarchical Generation of Molecular Graphs using Structural Motifs

[7] Gnnexplainer: Generating explanations for graph neural networks

[8] Motif-Driven Contrastive Learning of Graph Representations

6. Minor comments
	- For reproducibility, best performing hyperparameters could be mentioned.
	- The start of the sentence should be capitalized.


**Summary Of The Paper:**

The paper proposes a novel molecular graph representation learning method by constructing a heterogeneous motif graph. In this graph, molecules and motifs are considered as nodes, and thus forming a heterogeneous graph. Motifs are extracted manually and important motifs are selected by TF-IDF. Moreover, the paper demonstrates that a multi-task learning framework is beneficial, thanks to sharing motifs across different datasets.

**Summary Of The Review:**

Overall, I think this paper is at the borderline. Although the paper is among the first to use motif for molecular graph representation learning, the paper lacks sufficient literature survey. Moreover, it would have been better if the paper had had discussions about why motifs are helpful in a data-driven perspective.

---

> ### Author Response · Authors · 2021-11-16
> **Response to Reviewer 6PpB**
>
> We thank the reviewer for the positive review and helpful comments.
>
> **Concern 1:** The paper lacks a clear justification on whether two molecules sharing motifs imply special meaning.
> In other words, is sharing a motif a clear indication that two molecules share common properties?
> In addition, a motif-motif edge is generated if at least one atom is shared. Does sharing an atom imply anything chemically?
>
> **Answer 1:** Thanks for pointing this. The motivations of our work are based on scientific research results in biomedical fields. In particular, our work is motivated by the following point.
>
> Motifs are highly related to the functionalities of graphs [1]. For example, it is well known that carbon rings and NO2 groups tend to be mutagenic [2]. This observation or scientific outcome has served as a kind of ground truth for GNN explanation [3, 4, 5].
>
> In our proposed method, a motif-motif edge is constructed if two motifs share some atoms. This construction is based on the intuition that physically connected motifs have some chemical relations. Such intuitions are widely used, such as n-gram in natural language processing and convolution in computer vision.
>
> We have updated these details in the introduction section and related work section.
>
> [1] Alon, Uri. "Network motifs: theory and experimental approaches." Nature Reviews Genetics 8.6 (2007): 450-461.
>
> [2] Debnath, Asim Kumar, et al. "Structure-activity relationship of mutagenic aromatic and heteroaromatic nitro compounds. correlation with molecular orbital energies and hydrophobicity." Journal of medicinal chemistry 34.2 (1991): 786-797.
>
> [3] Ying, Rex, et al. "Gnnexplainer: Generating explanations for graph neural networks." Advances in neural information processing systems 32 (2019): 9240.
>
> [4] Luo, Dongsheng, et al. "Parameterized explainer for graph neural network." arXiv preprint arXiv:2011.04573 (2020).
>
> [5] Yuan, Hao, et al. "On explainability of graph neural networks via subgraph explorations." arXiv preprint arXiv:2102.05152 (2021).
>
> ***
>
> **Concern 2:** The exact message passing process needs to be described. For example, the sizes of the feature vectors of a motif node and a molecular node should be different, and how they are put together for message passing is not clear.
>
> **Answer 2:** We added two pseudocode describing details of our models and edge sampling in Appendix A and B. In our model, we set the sizes of the feature vectors of a motif node the same as the molecular node. We use one-hot encoding as a motif node feature vector. For molecular nodes, we use the bag-of-words method to generate a feature vector. For example, if a molecule has the first three out of five motifs. The feature vector of the molecule should be [1, 1, 1, 0, 0].
>
> ***
>
> **Concern 3:** Why are there blanks in Table 1 regarding "MUTAGENICITY."
>
> **Answer 3:** Thanks for pointing out this. Since some papers do not have the performance of Mutagenicity and PTC datasets, we use '-' to denote no result in the original papers. We added a clarification in the caption for this.
>
> ***
>
> **Concern 4:** Some concerns on edge sampler.
>
> **Answer 4:** We added pseudocode and give more details of edge sampling in Appendix B. Our sampler is sampling edges based on edge types. We are using BFS to explore and search edge types. Our heterogeneous motif graph has two kinds of edges and nodes which means for each molecular node, all of their neighbors are important motif nodes. Thus, to retain as much important information as possible, we generate all edges in the first hop, and from the second hop, we randomly sample a fixed number of edges from motif-motif edges.
>
> ***
>
> **Concern 5:** The literature survey is not sufficient.
>
> **Answer 5:** We added more literature into our related work section.
>
> ***
>
> **Concern 6:** Minor comments
> For reproducibility, best performing hyperparameters could be mentioned.
> The start of the sentence should be capitalized.
>
> **Answer 6:** We added hyperparameters for each dataset in Appendix D.

---

### Official Review · Reviewer_swsR · 2021-11-02

**Correctness:** 3
**Technical Novelty And Significance:** 2
**Empirical Novelty And Significance:** 2
**Recommendation:** 3
**Confidence:** 4

**Main Review:**

### Strengths
The paper proposes a method of constructing a heterogeneous graph based on motif, which can effectively utilize motif-based features.

### Weaknesses

The proposed method of the paper is interesting, but the baseline for comparison is too weak. For example, the two datasets of OGB shown in Table 2 compare the methods that are ranked very low on the OGB leaderboard, so it is difficult to prove the power of the method. At the same time, the paper did not give a detailed explanation on why GIN was chosen as the feature representation of the learning graph, and from the results in Table 1, GIN does not have obvious advantages compared with other methods. And when doing atom-level feature embedding learning, the edge feature (Bond Type, etc.) is not used, and the edge feature should have a certain degree of contribution to the prediction of molecular properties.


**Summary Of The Paper:**

Aiming at the feature representation learning problem of molecular graphs, this paper proposes a heterogeneous motif graph composed of motif nodes and molecular nodes to realize the information interaction between motifs and molecules. The paper uses graph neural networks to learn and combine atom-level and motif-level graph feature representations, so as to improve the representation of molecules.

**Summary Of The Review:**

The motif-based heterogeneous graph construction method proposed in this paper is interesting, but the experimental results are difficult to prove the effectiveness of the method. The network architecture is slightly simpler, the chemical bond information of the molecules is not used in the atom-level feature extraction. Therefore, I am not in favor of recommending this paper.

---

> ### Author Response · Authors · 2021-11-16
> **Response to Reviewer swsR**
>
> Thanks for your insightful review.
>
> **Concern 1:** The baseline for comparison is too weak.
>
> **Answer 1:** We have added CapsGNN, WEGL, and GraphNorm [1, 2, 3] in table 1 and PNA [4] in table 2. Compared to these four SOTA models, our method can significantly outperform previous models which demonstrate the effectiveness of the proposed model.
>
> [1] Xinyi, Zhang, and Lihui Chen. "Capsule graph neural network." International conference on learning representations. 2018.
>
> [2] Kolouri, Soheil, et al. "Wasserstein embedding for graph learning." arXiv preprint arXiv:2006.09430 (2020).
>
> [3] Cai, Tianle, et al. "Graphnorm: A principled approach to accelerating graph neural network training." International Conference on Machine Learning. PMLR, 2021.
>
> [4] Corso, Gabriele, et al. "Principal neighbourhood aggregation for graph nets." arXiv preprint arXiv:2004.05718 (2020).
>
> ***
>
> **Concern 2:** The paper did not give a detailed explanation on why GIN was chosen as the feature representation of the learning graph.
>
> **Answer 2:** In this work, we proposed a novel learning framework for molecular graphs, which is orthogonal to any graph deep learning method. Based on our heterogeneous motif graph, we can apply any GNN model to learn a motif embedding. We use GIN since it has been one of the most popular GNNs. But this choice is only for evaluation. We can apply any graph-based GNN in our framework. In table 2, we add more results for using PNA in our framework. The result shows that our motif-level embedding can bring additional motif-level information that benefits the generalizations of various existing GNNs.
>
> ***
>
> **Concern 3:** When doing atom-level feature embedding learning, the edge feature (BondType,  etc.)   is not used,  and the edge feature should have a certain degree of contribution to the prediction of molecular properties.
>
> **Answer 3:** There is a misunderstanding here. We used edge features in experiments on OGB datasets for atom-level representation learning. For small datasets in table 1, since most of the previous literature does not use edge features on these datasets, we did not use edge features for a fair comparison.

---

### Official Review · Reviewer_FtFN · 2021-11-03

**Correctness:** 4
**Technical Novelty And Significance:** 3
**Empirical Novelty And Significance:** 3
**Recommendation:** 5
**Confidence:** 4

**Main Review:**

The core strength of the paper is the strong experimental results. They obtained results that are much better than existing baselines on graph classifications. In terms of their ideas I find their TD-IDF approach natural, and I find the idea of taking into account molecular motifs very appealing and natural as well.

There are several concerns that I believe the authors should address for this paper to be of publishable quality:

1. One of my major concerns is that the authors did not provide a sufficiently detailed/rigorous description of the HM-GNN model/any of the related procedures like the edge sampling. The high level description of the model is based on very two short paragraphs in section 2.3, as well as a graphical illustration in figure 3. Lots of details of the model are left out/undescribed. I expect at the very least some pseudocode to describe some details, i.e. the neural network architecture/training, the initialization etc. The GNN for the atom level embedding is also glossed over. I believe that for publication in a venue like ICLR, the authors should at the very least give a more precise and detailed description of the model that they are proposing. There is only a very short appendix provided and many details (e.g. details on the proportion that they used for edge sampling etc)  are nowhere to be found unless the reader delve into their code.

2. Another one of my concerns is about the Edge sampling scheme. I have no doubts that it will actually speed things up (the complexity depends on the number of edges, so of course if we only select a subset of the edges the process is sped up). What the authors did not provide is any kind of justification, whether from a mathematical or from a molecular perspective, for why they picked this scheme/how they know their scheme isn't deleting important information? The authors propose to "randomly select some molecular nodes as starting nodes" (how many nodes? why random? Wouldn't some nodes be more important than others, especially in molecular settings?) and conduct a BFS where they sample a fixed portion of edges hop by hop (why fixed proportion? how does one pick the proportion?) I understand this is a heuristic that works well in practice but I expect a little more justification and details.

Overall, I will say that this is a paper with a lot of potential, and many of the ideas are likely to be useful in graph machine learning and molecular representation learning in general. However, the authors wrote the paper in a way where many of the core procedures are described in short paragraphs via words, and where lots of the important details, design decisions, hyperparameters etc are completely left out and are not sufficiently motivated/justified. Similarly ambiguous language is used in the experimental section. I would highly suggest the authors to include more precise descriptions and details in the paper, and to provide more justifications to their heuristics.

**Summary Of The Paper:**

This paper propose the Heterogeneous Motif Graph Neural Network (HM-GNN) which is based on a motif level graph representation that takes into account commonly occurring motifs like rings in molecules. They apply this to multi-task settings and obtained good experimental performances. They propose an edge sampling scheme to reduce the computational costs needed for training.

**Summary Of The Review:**

A paper with a lot of potential, but lots of important details are left out and lots of decisions are stated without sufficient justifications. As of right now, this paper is not in publishable state. After some revision however and some improvement in their exposition, I believe this can be a good paper.

---

> ### Author Response · Authors · 2021-11-16
> **Response to Reviewer FtFN**
>
> Thanks for your insightful review.
>
> **Concern 1:** Need a sufficiently detailed/rigorous description of the HM-GNN model/any of the related procedures like the edge sampling.
>
> **Answer 1:** Thank you for pointing out this. We have added two pseudocode and detailed explanations of our model and edge sampler to Appendix A and B.
>
> ***
>
> **Concern 2:** Why did they pick this scheme/how do they know their scheme isn't deleting important information?
>
> **Answer 2:** We have added more explanation of the edge sampler into section 2.5 and Appendix B.
> We are trying to retain important motif information by sampling edges based on edge type. Our heterogeneous motif graph has two kinds of nodes and edges. We can find that for all molecular nodes that need to be classified, all motifs connected to them are important. Thus, we aim to sample as many useful motif nodes as we can. To achieve it, we sample all edges in the first hop, and starting from the second hop, we randomly sample motif-motif edges.
>
> ***
>
> **Concern 3:** The authors propose to "randomly select some molecular nodes as starting nodes" (how many nodes? why random? Wouldn't some nodes be more important than others, especially in molecular settings?)
>
> **Answer 3:** In our proposed method, we construct a Heterogeneous Motif Graph by using each molecular graph as a node. This means each molecular node is a training example in a dataset. Since each training example is treated equally during training, the sampling process is random to ensure this equality. The number of start nodes is a hyper-parameter in the experiments since it needs to be determined by the availability of computational resources. Basically, the number of starting nodes works similar as the batch size when training a regular neural network.
>
> ***
>
> **Concern 4:** The authors propose to conduct a BFS where they sample a fixed portion of edges hop by hop (why fixed proportion? how does one pick the proportion?
>
> **Answer 4:** Thank you for pointing out this mistake. We have revised this to randomly sample a fixed size of edges. Sampling a fixed size of edges can help us control the size of the sampled graph by considering the actual availability of computational resources. We retain all first-hop edges to ensure effective learning of feature representations for motif nodes.

---

### Decision · Program_Chairs · 2022-01-20

**Decision:**

Reject

**Comment:**

The paper introduces a graph neural network for molecules which takes into account motif-level relationships. The paper received borderline reviews, with three reviewers voting for reject, and one for accept.  After the rebuttal, the reviewers did not change their scores. Overall, it seems that the paper has some merit, with good experimental results. Nevertheless, it suffers from two issues (i) the positioning with respect to other motif-based approaches is not clear enough, making the novelty hard to assess; (ii) there is a lot of room for improvement in terms of clarity. Therefore, the area chair follows the majority of the reviewers' recommendations and recommends a reject.